

# Diagnosing the AMOC slowdown in a coupled model: a cautionary tale

Justin Gérard[1] and Michel Crucifix[1]

[1]Université catholique de Louvain (UCLouvain), Earth and Life Institute (ELI), Louvain-la-Neuve, Belgium

**Correspondence:** Justin Gérard (justin.gerard@uclouvain.be)

**Abstract.** It is now established that the increase in atmospheric $CO_2$ is likely to cause a weakening, or perhaps a collapse of the Atlantic Meridional Overturning Circulation (AMOC). To investigate the mechanisms of this response in CMIP5 models, Levang and Schmitt (2020) have estimated offline the geostrophic streamfunction in these models and decomposed the simulated changes into a contribution caused by the variations in temperature and salinity. They concluded that under a warming scenario, and for most models, the weakening of the AMOC is fundamentally driven by temperature anomalies while freshwater forcing actually acts to stabilize it. However, given that both 3-D fields of ocean temperature and salinity are expected to respond to a forcing at the ocean surface, it is unclear to what extent the diagnostic is informative about the nature of the forcing. To clarify this question, we used the Earth system Model of Intermediate Complexity (EMIC) cGENIE, which is equipped with the C-GOLDSTEIN friction-geostrophic model. First, we reproduced the experiments simulating the RCP8.5 warming scenario and observed that cGENIE behaves similarly to the majority of the CMIP5 models considered by Levang and Schmitt (2020), with the response dominated by the changes in the thermal structure of the ocean. Next, we considered hysteresis experiments associated with (1) water hosing and (2) $CO_2$ increase and decrease. In all experiments, initial changes in the ocean streamfunction appear to be primarily caused by the changes in the temperature distribution, with variations in the 3-D distribution of salinity compensating only partly for the temperature contribution. These experiments also reveal limited sensitivity to changes in the ocean's salinity inventory. That the diagnostics behave similarly in $CO_2$ and freshwater forcing scenarios suggests that the output of the diagnostic proposed in Levang and Schmitt (2020) is mainly determined by the internal structure of the ocean circulation, rather than by the forcing applied to it. Our results illustrate the difficulty of inferring any information about the applied forcing from the thermal wind diagnostic and raise questions about the feasibility of designing a diagnostic or experiment that could identify which aspect of the forcing (thermal or haline) is driving the weakening of the AMOC.

## 1 Introduction

A commonly used illustration for the global oceanic circulation is that of a 'conveyor belt' driven by both wind stress and buoyancy fluxes at the ocean-atmosphere interface (Dijkstra, 2005). The present study focuses on the thermohaline part of this circulation and more specifically on the Atlantic Meridional Overturning Circulation (AMOC). The AMOC emerges as the result of three contributions, which are (1) the preconditioning of salty water mass and its horizontal northward advection,



(2) the buoyancy loss due to exchanges with the atmosphere and (3) the density gradient across the water column resulting from convection. This convection occurs primarily at subpolar latitudes in the Labrador and Greenland Seas and maintains the large-scale circulation via the geostrophic balance between the pressure and Coriolis forces (Weijer et al., 2019). It is widely known (e.g. Dijkstra (2005)) that the atmospheric buoyancy fluxes have opposite contributions to the AMOC: a purely thermal circulation would be a clockwise flow that follows the sinking of colder water in the North Atlantic, while a strictly haline circulation would be an anti-clockwise flow due to the sinking of saltier water at the equator. The current balance has water masses sinking to depths of 2-3 km forming the North Atlantic Deep Water (NADW), which propagates southward until it reaches the surface again (Rhein et al., 2017). About 80% of the ocean deep water returns to the surface in the Southern Ocean, while the remaining portion returns to the sea surface via upwelling to the thermocline at low latitudes (Tamsitt et al., 2017). It is now well established that the AMOC is a crucial part of the climate system, due to its essential contribution to the global circulation, which itself acts as a redistribution system that transports large amounts of heat and salt throughout the Northern Hemisphere (Jungclaus et al., 2006). Even if the current AMOC slowdown remains an open question, models predict a substantial AMOC slowdown if atmospheric $CO_2$ concentrations continue to increase Caesar et al. (2021); Kilbourne et al. (2022); Latif et al. (2022); Zhu et al. (2023). A reduction in the AMOC would cause an equatorward shift of the Intertropical Convergence Zone (ITCZ), a weakening of the Asian and Indian summer monsoons and a cooling of the Arctic (Buckley and Marshall, 2016).

The potential triggers and mechanisms for changes, possibly abrupt, in the AMOC and the associated impacts on the climate system are still an active area of research (Mehling et al., 2022). Specifically, the possibility that the thermohaline circulation (THC) could exhibit bistability and hysteresis (quasi-irreversibility) was suggested more than 60 years ago (Stommel, 1961; Rahmstorf, 1996). Bistability has since been identified and studied across a wide range of models though not all models exhibit this feature with the present boundary conditions (Rahmstorf et al., 2005; Hawkins et al., 2011; Weijer et al., 2019; Jackson et al., 2022). For the AMOC, the different states would correspond to vigorous versus weak overturning and are often referred to as 'on' vs 'off' states, respectively (Jackson et al., 2017). It is not excluded that with global warming, the AMOC will end up locked up in an 'off' state, but a more likely scenario is that the AMOC will undergo a transient decay and partial recovery (Jackson et al., 2022; Weijer et al., 2019).

As a general rule, anomalies in temperature and freshwater input, particularly close to the convection zone, can decrease ocean density and weaken the circulation. Then, different feedbacks are crucial in determining how and when the circulation may be brought to a potential collapse. On the one hand, this weakening decreases heat transport to the North, thereby increasing ocean density in a negative feedback loop. On the other hand, the decrease in salt transport amplifies the initial perturbation in a positive feedback loop, called the advective salinity feedback. The response of the atmosphere may further interact with these feedbacks, by modulating heat and freshwater exchanges. However, as the ocean salinity does not feedback directly on precipitation, salinity anomalies are able to propagate freely and are decisive in amplifying the perturbation via the advection feedback (Dijkstra, 2005).

To explore the circulation response and the related mechanisms, authors usually choose to force the system either with freshwater fluxes or atmospheric $CO_2$ concentration. What value these forcings need to reach to induce a complete shutdown





of the circulation depends on the model (Liu et al., 2017) and on their rate of increase (Stocker and Schmittner, 1997). In many models, when the atmospheric concentration of $CO_2$ increases from preindustrial, it causes a weakening or even a collapse of the AMOC mainly through the reduction of the vertical density gradient (Thorpe et al., 2001). To investigate some mechanisms of the slowdown, Hirschi and Marotzke (2007) developed a diagnostic based on the thermal wind relationship. Specifically, they reconstruct the overturning streamfunction using the thermal wind balance. Then, fixing either the temperature or the salinity field to the initial conditions of a transient experiment allows us to examine the haline and thermal contributions to the variations in the overturning streamfunction. Levang and Schmitt (2020) applied this diagnostic to analyse the output of 26 models of the CMIP5 experiment, in response to the the representative concentration pathway 8.5 (RCP8.5) scenario. Overall, they concluded that temperature dynamics are responsible for AMOC weakening, while freshwater forcing instead acts to strengthen the circulation.

The objective of the present study is twofold. First, we apply the same diagnostic as Levang and Schmitt (2020) to the cGENIE framework which contains the efficient C-GOLDSTEIN climate model (Marsh et al., 2011). Because this model is used in many paleo and future climate studies, we want to know where this model stands with respect to the CMIP5-standard models. In this respect, the present study is a contribution towards the documentation of a model widely used in palaeoclimate research. For example, it was used to explore past ocean acidification by illustrating the variation of ocean carbonate properties under different atmospheric $CO_2$ perturbations (Hönisch et al., 2012), reconstruct the carbon-cycle dynamics during the Palaeocene–Eocene thermal maximum event (Gutjahr et al., 2017) and investigate the Pacific meridional overturning circulation during the last glacial maximum (Rae et al., 2020). Second, due to its lower complexity level, cGENIE allows us to do long experiments at a relatively small cost. It implies that we will be able to examine, with the Levang and Schmitt (2020) diagnostic, how the thermal and haline contributions unfold until several centuries/millennia after the collapse when the ocean salinity and temperature fields have reached equilibrium. Then, by comparing the output of experiments with $CO_2$ and freshwater forcing, we may critically examine what this diagnostic approach tells us about the nature of the forcing.

## 2 Methods

### 2.1 Model description

cGENIE is an Earth system model of intermediate complexity in the definition given by Claussen et al. (2002). The framework of cGENIE consists of an ocean circulation model (3D), a dynamic-thermodynamic sea ice model (2D) and an atmospheric energy moisture balance model (2D). The ocean model accounts for the horizontal and vertical transport of heat, salinity and biogeochemical tracers. The circulation is simulated through advection, convection, and mixing. The sea ice model is similar to that employed in the UVic model of Weaver et al. (2001) and consists of a simple representation of sea ice thermodynamics following Semtner Jr (1976) and Hibler (1979). Specifically, the temporal evolution of sea ice thickness is dictated by the thermal balance, while the dynamical equations are solved for the fraction of the ocean surface covered by sea ice in any given region and the average height of sea ice (Marsh et al., 2002). Precipitation is assumed to go directly into the ocean ignoring the ice, and sublimated water from sea ice is directly added to the atmosphere. The atmosphere energy moisture balance model





is a one-layer, two-dimensional model that includes the horizontal transport of heat and humidity in the atmosphere through
advection and diffusion by prescribed winds and mixing. Surface albedo and wind stress are constant for the entirety of a given
simulation. In its current version, the model does not provide a dynamic hydrological scheme for the continent. Evaporation
on land is set to zero and precipitation is added to coastal cells according to a runoff map. Atmospheric precipitation is then
computed by instantly removing any excess humidity beyond a certain relative humidity threshold. A complete description of
these models is provided in Edwards and Marsh (2005); Marsh et al. (2011). In addition, cGENIE also includes modules for
biogeochemical tracers in the ocean BIOGEM (3D) and for the atmosphere chemistry ATCHEM (2D) (Lenton et al., 2007;
Ridgwell et al., 2007). These modules are essential in most applications of cGENIE but are not relevant here.

In our experiments, the ocean is on a $36 \times 36$ horizontal equal-area grid with 16 logarithmically spaced levels along the
vertical (from 80 m to 765 m). The equal-area feature of the horizontal grid results from the uniform spacing in the longitude
and sine of the latitude. The continental configuration and the boundary conditions are the ones used in Crichton et al. (2021)
which correspond to a preindustrial climate (atmospheric $CO_2$ concentration of 280 ppm), while all the ocean and atmosphere
parameters are taken from Cao et al. (2009). We use the originally prescribed moisture transport from the Atlantic to the Pacific
(as the AMOC is not able to sustain without it in cGENIE), and modify atmospheric diffusivity over the Southern Ocean and
Antarctica as well as the sea ice diffusivity parameter — as in Ödalen et al. (2020); Crichton et al. (2021) — to improve the
seasonal sea-ice properties in the Southern Ocean. The seasonal cycle of solar radiation is activated as well as the climate
response to changes in atmospheric $CO_2$. To obtain a configuration corresponding to a preindustrial climate equilibrium an
extended simulation of 30000 years is conducted using an atmospheric $CO_2$ concentration of 280 ppm. This state provides the
initial condition for both sets of experiments (RCP8.5 and hysteresis) presented below.

## 2.2 Geostrophic approximation

Since the AMOC is nearly geostrophically balanced at the interannual time scale, the transport can be determined using thermal
wind and density profiles obtained from the zonal boundaries of the basin, as reported in Buckley and Marshall (2016). The
diagnostic presented in Levang and Schmitt (2020) uses this thermal wind balance to build a geostrophic streamfunction
to approximate the one computed by the CMIP5 models. In their study, they showed that for all the models considered, this
approximation is good especially when it comes to estimating the maximum of the streamfunction. To compute the geostrophic
streamfunction, we first need the expression of the meridional velocity $v_g$ as a function of depth. It is obtained from the thermal
wind balance:

$$v_g(z) = \frac{g}{\rho_0 f} \int_{-H}^{z} \frac{1}{L(z)} (\rho_w - \rho_e) \mathrm{d}z, \qquad (1)$$

with $z$, the depth variable verifying $-H \le z \le 0$; $g$, the gravitational acceleration; $\rho_0$, the reference density; $f$, the Coriolis
parameter; $H$: as the ocean depth; $L(z)$, the basin width; $\rho_w$ and $\rho_e$ the density on the western and eastern margins, respectively.
The density fields used for the computation of the meridional velocity are either directly used from the model output (if





available) or generated offline from the temperature and salinity field using the state equation of density. In cGENIE, this state equation is a third-order polynomial approximation of the UNESCO equation of state (Winton and Sarachik, 1993). The expression of the geostrophic streamfunction is then obtained by integration of the meridional velocity

$$\Psi_{\mathrm{geo}}(z) = \int\limits_{x_w}^{x_e} \int\limits_{-H}^{z} \left( v_g(z) - \frac{1}{H} \int\limits_{-H}^{0} v_g(z)\mathrm{d}z \right) \mathrm{d}z\mathrm{d}x, \tag{2}$$

where mass conservation is achieved by removing the depth average velocity. This allows the streamfunction to fulfil the
130 condition $\Psi_{geo}(z) = 0$ for $z = 0, -H$. In the North, the Atlantic basin is open through the Bering Strait, implying water exchanges of the order of about 1 Sv. These exchanges are not included in our approximation because they are relatively small compared to the total overturning. The full development of the thermal wind diagnostic and the related geostrophic streamfunction can be found in Hirschi and Marotzke (2007).

## 3  Results

### 135  3.1  RCP8.5

cGENIE is forced using the atmospheric $CO_2$ concentrations of the RCP8.5 scenario used in the CMIP5 ensemble. Our RCP8.5 simulation runs from a preindustrial climate to the year 3000, but the year 2000 acts as the initial condition of the results presented here. This setup allows for realistic behaviour and direct comparison with the work of Levang and Schmitt (2020).

Figure 1a shows the overturning stream function associated with the AMOC, in Sv ($1 \text{ Sv} \equiv 10^6 \text{ m}^3\text{s}^{-1}$) for the year 2000.
The streamfunction in cGENIE is weaker than in the multimodel mean, hardly reaching a maximum of 13 Sv against 20 Sv for the CMIP5 multimodel mean. While the difference is significant, it is not a strong cause for concern as climate models show great variability in the reconstruction of the overturning strength. The CMIP5 models also have maximum AMOC values ranging from 10 to 30 Sv around the year 2000, with cGENIE falling into that range. As already mentioned, the geostrophic approximation proves to be a reliable method for estimating the streamfunction maximum, which is located a bit South of $54°$
N (specifically at $53.66°$ N) in cGENIE. Figure 1b illustrates the spatial pattern of the AMOC following 1000 years of RCP8.5 scenario, while Figure 1c shows the time series of different streamfunctions maximum at a latitude of $54°$ N. The AMOC has completely collapsed by the year 3000, with Antarctic waters having filled most of the Atlantic basin. The collapsed character of the AMOC can be further confirmed by inspecting the dissolved oxygen profile of the Atlantic, known as a reliable proxy for water ages (not shown here) (Marshall and Plumb, 2013).

The collapse of the AMOC under the RCP8.5 scenario in cGENIE did not surprise us. In preparatory work, we observed that the AMOC collapses between $CO_2$ concentrations of 5 and 6 times Preindustrial Atmospheric Level (PAL) when $CO_2$ concentration slowly increases (specifically, 5000 years to move from $n$ to $n + 1$ times PAL). However, we also found that a collapse of the AMOC occurs for lower $CO_2$ concentration if the increase rate is higher, as is the case in this experiment. Rate-dependent tipping of the AMOC is a well-established phenomenon, already described by Stocker and Schmittner (1997)



and that has been shown and analysed more recently in a global ocean model (Lohmann and Ditlevsen, 2021). The RCP8.5
scenario happens to trigger this rate-induced tipping.

    To investigate the respective contributions of temperature and salinity, two partial geostrophic streamfunctions are derived
offline from density fields. Similarly to Levang and Schmitt (2020), these fields are generated using the temperature or salinity
evolution, only, keeping the other variable equal to its initial distribution. We find that, throughout the experiment, temperature

and salinity fields alternate their respective contributions to the geostrophic streamfunction anomaly. Specifically, three distinct
phases emerge, taking place between 2000-2100, 2100-2300, and 2300-3000 years, respectively (Figure 1c). The phases are
bounded by changes in the concavity of the two partial geostrophic streamfunctions (the blue and red curves in Figure 1c).
The geostrophic streamfunction decreases throughout the simulation and this evolution results from partially compensating
contributions of temperature and salinity, which tend to act against each other.

During the first 100 years of the simulation (phase 1), the cGENIE and geostrophic streamfunctions (respectively the dotted
and solid black curves in Figure 1c) are very close to each other, attesting to the adequacy of the geostrophic approximation.
This is not surprising, considering that C-GOLDSTEIN, the ocean circulation model, uses a friction-geostrophic approximation
to estimate the circulation. Furthermore, the behaviour of the diagnostic obtained is very similar to the multimodel mean made
by Levang and Schmitt (2020) (see Figure A1). If we follow Levang and Schmitt (2020), the results suggest that the thermal

component of the circulation induces a significant weakening while the haline component acts to strengthen the streamfunc-
tion, the net effect being a decrease in the AMOC strength. However, we also observe that during the next 200 years (phase
2), the respective contributions of temperature and salinity on the overturning circulation reverse their trends. The thermal
forcing acts to increase the geostrophic streamfunction, and the haline forcing causes a decrease. For the remaining part of the
simulation (phase 3), the trend reverses again with the temperature and salinity fields regaining their initial contribution as for

the first 100 years of the experiment. These trend inversions have not been observed by Levang and Schmitt (2020) because
the corresponding CMIP5 experiments were too short to evidence it.

    We now provide a more in-depth analysis of the behaviour of the geostrophic function components. In equation 2, the
overturning streamfunction is a vertical integral of the zonal density gradient —following the principle of the thermal wind
balance— down to the ocean sea floor. Consequently, the zonally integrated, geostrophic streamfunction depends on the values

of density on the western and eastern sides, only. In the initial condition, the thermal contribution is positive and dominates
that of the salinity, which is negative, justifying the statement that the streamfunction is mainly thermally driven (see Figure
A2).

    During the first phase of the simulation (year 2000 to 2100), the increase in surface temperature reduces the intensity of
deep-sea mixing at convection sites, in the North-eastern Atlantic. On the western side of the ocean, the temperature increases

along most of the water column because the heat is no longer effectively evacuated by the convection process. On the eastern
side, where convection does not normally occur in this model, the combined action of the atmospheric forcing (positive) and
the reduced vertical ventilation in the west cause only moderate warming. Thus, the western margin experiences a greater
warming trend than the eastern one, leading to a reduction in the zonal density gradient across nearly the entire water column,
which in turn weakens the streamfunction (see Figure 2a). The warming of the western margin has previously been identified



as the leading cause of AMOC weakening Levang and Schmitt (2020). This effect is partly compensated for by a reduction of the east-west salinity gradient which, as we recall it, tends to act against the overturning streamfunction (see Figure A2b and d). Indeed, the equatorial advection of salt propagates less extensively towards the east, depriving the eastern boundary of a source of salt, while leaving the west almost unchanged. This causes a reduction in the density along the first 1500 m of the eastern basin margin, thereby reinforcing the geostrophic overturning (see Figure 2d). Again, because density variations

induced by temperature perturbations are larger than those resulting from salinity, the overall modification of density induces a weakening of the streamfunction, justifying the driving role played by temperature during this phase.

As the intensity of the overturning decreases, less buoyant water is being advected from the equatorial regions to the North, via the western boundary current. This prepares the second phase (year 2100 to 2300), which is marked by a reduction and southward shift of the overturning cell, depriving the first 1500 meters of the western boundary of a source of warm and salty

water. On average, the western boundary experiences lower warming compared to the east but it loses more salinity (see Figure 2b and e), and this further reduces the zonal density gradient (and doing so the streamfunction). Below 1500 m, the thermal and haline fields homogenize due to the decrease in the depth of the AMOC. The salinity gradient at depth is fairly close to what it was during the first phase of the experiment. However, as the western boundary is no longer supplied with warm water its temperature decreases compared to the year 2100; it cools and this is now the temperature that partially compensates for the

salinity effect that drives the slowdown of the AMOC.

During the final phase (year 2300 to 3000), the driving mechanism shifts towards mixing and contamination of lower ocean layers by Antarctic waters, which follows the complete shutdown of deep convection. Although deep circulation stops, the subtropical gyre continues to operate and allows for relatively warm and saline waters to propagate northward along the western boundary (see Figure 2c and f). These inputs cause the first 750 meters (mainly through eddy-diffusion) of the western

boundary to be warmer and saltier than on the eastern boundary, maintaining the horizontal density differences at these depths. This explains why the geostrophic streamfunction in Figure 1c never properly reaches 0 but rather converges to a value around 3 Sv. For the rest of the water column, the contamination of Antarctic Bottom Water and mixing homogenize completely the thermal and haline fields, making their contribution to the zonal density gradient negligible. As a result, during this final phase, temperature changes drive, again, the weakening of the streamfunction with the remaining zonal density gradient being only

maintained by the surface circulation.

We also observe that the quality of the geographic approximation deteriorates around year 150, compared to the initial condition, although it stays decent for the entire run. We attribute this deterioration to the latitude shift of the streamfunction maximum when the AMOC weakens. Because the $CO_2$ increase forces the overturning cell to move southward, one could apply the thermal wind diagnostic while staying at the latitude of the streamfunction maximum (see Figure 1d). This method seems

more relevant, but tracking the maximum of the overturning streamfunction implies changing the latitude of the diagnostic as the overturning cell moves. Due to the coarse spatial resolution of cGENIE, each of the abrupt jumps in the geostrophic curves corresponds to a change in the latitude of the streamfunction maximum. We found, however, that south of 30° N, as the Coriolis factor is smaller and the thermal wind approximation becomes more questionable, the difference between the true value and the geostrophic approximation could reach about 10 Sv. Whether using the geostrophic diagnostic at a fixed latitude




of 54° N or tracking the latitude as the overturning streamfunction shifts do not qualitatively affect the results discussed so far (curves shown on 1d are very similar to those shown on 1c). One clearly distinguishes the same three phases along the decrease in the maximum overturning streamfunction. This sensitivity analysis to the diagnostics confirms that the three phases that we have identified are not an artefact of using a fixed latitude (54° N), but rather a more widespread feature throughout the North Atlantic. However, one has to note that the timing and demarcation of these three phases can differ from one latitude to the

other.

## 3.2 The nature of the forcing

Having established the underlying mechanisms that lead to the weakening of the AMOC in cGENIE, our next goal is to investigate the information that can be inferred about the nature of the forcing applied to the circulation using the same diagnostic. To reach this goal, two different hysteresis experiments are conducted: one using $CO_2$ and the other freshwater.

Simulations involve a linear increase and subsequent decrease (symmetric in time) of the forcing value over a period of 40000 and 20000 years, respectively (see Figure 3). In the $CO_2$ experiment, the concentration of $CO_2$ is incrementally increased from 1 to 10 times the preindustrial concentration in 18000 years after staying constant for 2000 years. Simulations show that, in cGENIE, the system has to stay at low atmospheric $CO_2$ concentrations for a few hundreds of years to allow the circulation to recover. The freshwater hosing experiment is identical to the one realised in Rahmstorf et al. (2005). It involves the addition

of freshwater in a uniform manner to the latitude band of 20-50° N across the Atlantic with a rate of change of 0.05 Sv per 1000 model years. Targeting these latitudes is driven by the motivation to avoid direct forcing of the high-latitude convection regions. With this framework, the model is forced from -0.2 Sv to 0.3 Sv in 10000 years. The slow variation of forcing rates keeps the system in a quasi-equilibrium state, which is crucial for hysteresis experiments.

Figures 4a and b show the results for experiments of hysteresis forced by atmospheric $CO_2$ concentration and freshwater

flux, respectively. Figure 4a includes two curves, a solid one representing the maximum streamfunction at 54° N and the dotted one showing the same maximum but considering all latitudes above 30° N. As for the RCP8.5, only the curve at a fixed latitude will be studied as it differs very little from the second one, except for a more abrupt recovery of the AMOC and a slight underestimation of the streamfunction value when it is low. The second curve is used as further verification of the robustness of the diagnostic already tested in section 3.2. Figure 4b also includes two curves, the solid one corresponds to

the maximum streamfunction at 54° N with a total ocean salt inventory kept constant, and the dotted one represents the same curve but with an open salt inventory. To achieve a constant salinity inventory in the experiment, the freshwater forcing applied across the Atlantic is compensated for by applying the exact opposite forcing but distributed over the Pacific. Overall, the two curves show close similarities providing evidence that the open/close aspect of the total ocean salt inventory is not of major importance in this experiment. While this result has already been previously observed in the original study (Rahmstorf et al.,

2005), it serves as a verification of our experimental design and the robustness of our findings.

Figures 4c and d represent the thermal wind diagnostic of the hysteresis experiment with $CO_2$ and freshwater respectively. They show the time evolution of the same quantities as in Figure 1c at a latitude of 54° N. As previously mentioned, the geostrophic approximation allows for proper reconstruction of the streamfunction in cGENIE. Similarly, the hysteresis ex-





periment with $CO_2$ exhibits the presence of the three phases identified in the RCP8.5 experiment (see Figure 4c). The first phase (from 1 to 3 times PAL), where the streamfunction slowdown is driven by changes in thermal structure mainly due to the weakening of the convection. The second phase (from 3 to 5.5 times PAL) is characterized by an equatorward shift of the overturning cell, which generates a predominance of the salinity signal in the weakening of the overturning circulation as the northward advection of warm and salty waters diminishes. The final phase (from 5.5 to 10 times PAL) is governed by the mixing and contamination of Antarctic bottom waters, which reverses the trend one last time, with a complete stoppage of the AMOC that is maintained by the thermal field. At the end of the simulation, an overshoot is observed. The overshoot at AMOC recovery is often observed in models (Rahmstorf et al., 2005) and is a transient effect related to the accumulation of buoyancy in the 'off' state of the circulation. Here, we find that this phenomenon is mainly thermally driven, with the salinity gradient working against the temperature gradient. The fact that this hysteresis experiment with $CO_2$ displays close similarities to the RCP8.5 is not very surprising, except for the presence of a saw-tooth-shaped event around 4 PAL that we will explore later on.

Perhaps more surprising is that the hysteresis experiment with freshwater flux also exhibits very similar behaviour to that of the RCP8.5 (see Figure 4d), because we know that in the latter case, the forcing is freshwater and not temperature. We are not the first to find that freshwater hosing is able to reduce the strength of the AMOC through temperature effects as similar findings have already been achieved by Haskins et al. (2020). Again, we see three phases (the second one is a little bit shorter), with the trends overall dominated by the changes in the thermal structure of the ocean and with salinity taking the lead for the actual shutdown of the circulation. Having a similar decomposition and behaviour of the temperature and salinity changes in experiments with two clearly different forcings ($CO_2$ in Figure 4a and freshwater in Figure 4c) comes as a warning that the thermal wind diagnostic is indicative of the inner structure and response of the AMOC, but it does not inform us about the nature forcing applied to it. Therefore, only using this offline diagnostic does not allow for separating the respective contributions of temperature and freshwater flux changes on the weakening of the AMOC that comes with the increase of atmospheric $CO_2$ concentration.

Now we return to the 'saw-tooth-shaped' event visible on both curves of Figure 4a. We find that this 'saw-tooth' behaviour is a furtive occurrence of an oscillation that is triggered as the system evolves in quasi-equilibrium under particular $CO_2$ concentrations. The oscillation regime is also triggered in the freshwater forcing hysteresis experiment, but the oscillation is weaker and only appears when the freshwater flux forcing in the Atlantic is compensated for in the Pacific (see zoom in Figure 4b). The occurrence of natural oscillations in the AMOC on the multicentennial/millennial timescale is a familiar phenomenon that has been observed across many different models (Sakai and Peltier, 1997; Thornalley et al., 2009; Peltier and Vettoretti, 2014; Sévellec and Fedorov, 2014; Li and Yang, 2022; Vettoretti et al., 2022). In cGENIE, some millennial oscillations have been highlighted by Keane et al. (2022) for different ranges of atmospheric $CO_2$ concentrations and an idealized continental configuration. Oscillations in the THC have also been observed in cGENIE as freshwater forcing is applied to the system when close to a bifurcation point (Lenton et al., 2009). To prove that it is possible to lock the system in a stable oscillatory mode, additional simulations were performed. These simulations involve the linear modification of the forcing during half the duration of the experience until it reaches a threshold value for the rest of the run (1170 ppm for $CO_2$ and 0.06 Sv for freshwater). Figure 5 shows the stable oscillations obtained when the system is forced with $CO_2$ (left) and freshwater (right). From this, we can




conclude that for a given $CO_2$ concentration or freshwater forcing value it is possible to induce self-sustained oscillations.
Self-sustained oscillations in the ocean can have different origins and a full investigation of the mechanism of their specific occurrence here is beyond the scope of the present study. We however note a few facts:

- Quite trivially, we get more of these oscillations if the forcing change is slower since more time is spent in the window of forcing compatible with the oscillations (see Figure 5).

- The oscillation occurs during the phase where the evolution of the streamfunction is governed by the salinity field, and we find that it is driven by the haline contribution (see Figure4c).

- Simulations show that the oscillations from the freshwater forcing hysteresis experiment have a shorter period of typically a hundred years (see Figure 5b).

## 4 Discussion and conclusions

This study builds on the work of Levang and Schmitt (2020), who used the thermal wind diagnostic and various general circulation models to examine the AMOC weakening under the RCP8.5 warming scenario. Here, we used cGENIE, first to get some further confidence about the representation and sensitivity of the AMOC in this model widely used for palaeoclimate studies, and also because we wanted to understand better the value of the diagnostic itself. We established close similarities with the CMIP5 models during the corresponding time interval of the RCP8.5 scenario. We also highlighted the three-phase structure of the circulation collapse scenario, with first, the weakening of the convective mixing driving the thermal zonal gradient over the first 1500 m of the water column, across the North Atlantic; then, the northward advective weakening via the western boundary current drives a decrease in the salinity gradient; and finally, the collapse and invasion of much of the Atlantic by Antarctic-origin waters, with a surface density gradient persisting mainly through the surface gyre circulation. The three-phase structure is robust across different experimental setups, whether the primary driver is $CO_2$ increase or freshwater forcing. We found that the interplay between temperature and salinity is crucial for the shutdown of the circulation, with the initial slowdown being thermally driven and the salinity taking the lead for the termination of the overturning. We also tested the robustness of the implementation details of the diagnostic. Specifically, tracking the latitude of the maximum of the overturning stream function or staying localised at 54°N generates the same conclusions.

These results invite us to suggest a caveat with respect to Levang and Schmitt (2020) following conclusions: *Therefore, in CMIP5, temperature dynamics are responsible for AMOC weakening, while freshwater forcing instead acts to strengthen the circulation in the net. These results indicate that past modelling studies of AMOC weakening, which rely on freshwater hosing in the subpolar gyre, may not be directly applicable to a more complex warming scenario.* In this paper, we showed that the thermal wind diagnostic does not allow us to conclude whether variations in the AMOC produced under an increase of $CO_2$ are mainly driven by freshwater or heat flux exchanges with the atmosphere. In our simulations, even a pure freshwater forcing induces a slowdown of the AMOC driven by the temperature field and similar results were also found by Haskins et al. (2020) using two distinct general climate models. This lack of information is a consequence of running simulations where neither the



temperature nor salinity field is kept constant, meaning that the temperature field is influenced by circulation modifications induced by changes in the salinity field and vice versa. As a result, the signal of the perturbation forcing becomes increasingly weaker over the course of the simulation before disappearing completely from the offline thermal wind diagnostic. On the other hand, we agree with Levang and Schmitt (2020) on the relevance of using $CO_2$ concentrations to constrain the AMOC.

While the large majority of studies showcase the hysteresis behaviour of the AMOC using only a freshwater input, we manage to show that it can also be obtained by considering a $CO_2$ forcing. Interestingly, the hysteresis curve forced by $CO_2$ has revealed features that were less obvious with the hosing experiment, highlighting the potential benefits of considering different approaches when forcing the AMOC.

       The present findings contribute to the overall discussion about the suitability of EMICs, such as cGENIE, to study the

response of the large-scale thermohaline circulation. In the past, the cGENIE framework has proven to be a relevant choice for analyzing large-scale thermohaline circulation (Rahmstorf et al., 2005; Lunt et al., 2006; Lenton et al., 2007; Cao et al., 2009; Marsh et al., 2013; Keane et al., 2022). These findings also provide additional evidence that cGENIE can exhibit natural stable oscillations on the millennial time scales for a modern continental configuration whether the system is forced with $CO_2$ or freshwater.

However, there are some well-known limitations associated with the model that need to be taken into consideration, such as its coarse spatial resolution, a simplistic representation of the continents and the associated hydrological scheme, the absence of ice sheets and the lack of a dynamic atmosphere. Specifically, the atmospheric feedback in cGENIE is fairly elementary, with precipitation not responding as it would in a full general circulation model. For example, wind stress does not respond to climate change, and yet, this response could have different impacts on the gyre circulation and on the deep-water mixing, both in the

southern and the northern hemispheres. Furthermore, the coarse spatial resolution can induce an inadequate representation of circulation dynamics, particularly at high latitudes. The simplistic hydrological scheme and the absence of ice sheets lead to flaws in freshwater fluxes along coastal margins which can in turn influence the overturning.

       Despite all these drawbacks, the model is able to reproduce the general trend of the multimodel mean used by Levang and Schmitt (2020) and the simplifications that are made allow for long-term simulations with a much more reasonable computation

time. In this context, the similarity between the diagnostic scenario between the first phase of the collapse scenario in cGENIE, and that found in CMIP-class models Levang and Schmitt (2020) has proved to be fairly reassuring. We are in a position to suggest that the three-phase development of the AMOC collapse that we have found here could also occur in CMIP-class models. To our knowledge, this has not been tested so far, and we would naturally encourage such investigation to document similarities and potential differences between this EMIC and finer models.

*Code availability.* The code for the version of the 'muffin' release of the cGENIE Earth system model used in this paper, is tagged as v0.9.28, and is assigned a DOI: 10.5281/zenodo.5500719. Configuration files for the specific experiments presented in the paper can be found in the directory: genie-userconfigs/MS/gerard.crucifix.2023 (https://github.com/mcrucifix/cgenie.muffin). Details of the experiments, plus the command line needed to run each one, are given in the 'readme.txt' file in that directory. All other configuration files and boundary



conditions are provided as part of the code release. A manual detailing code installation, basic model configuration, tutorials covering various
aspects of model configuration, experimental design, and output, plus the processing of results, is assigned a DOI: 10.5281/zenodo.5500696.
All the different files will be dropped on Zenodo if the paper is accepted.

*Data availability.*  The data used for the RCP8.5 emission scenario were taken from http://www.pik-potsdam.de/~mmalte/rcps/.

*Author contributions.*  J.G conducted the cGENIE experiments, developed the code for the diagnostics, performed all computations, and
drafted the manuscript. Both authors contributed to the study design, discussion of results, writing the manuscript and approved the final
version.

*Competing interests.*  M.C. is a member of the editorial board of Earth System Dynamics.

*Acknowledgements.*  This research is funded by the Belgian National Fund of Scientific Research (FNRS), project WarmAnoxia, contract
PDR 40008052. Computational resources have been provided by the supercomputing facilities of the Université catholique de Louvain
(CISM/UCL) and the Consortium des Équipements de Calcul Intensif en Fédération Wallonie Bruxelles (CÉCI). We thank Professor Thierry
Fichefet for initiating the scientific question treated in this paper.



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



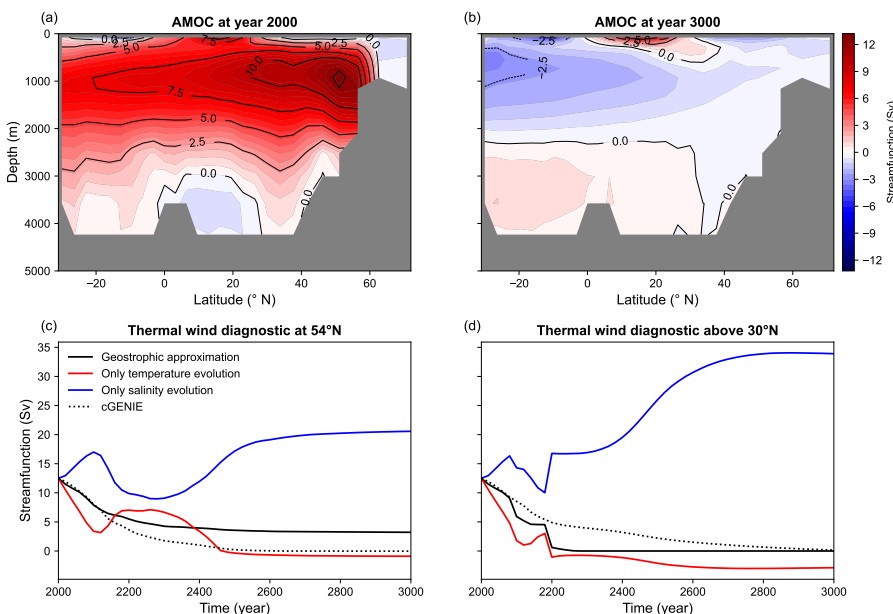

**Figure 1.** Measure of the AMOC strength (Sv) using the cGENIE streamfunction as a function of latitude and depth under the RCP8.5 warming scenario, corresponding to the year (a) 2000 and (b) 3000. Bold flow lines are spaced by 2.5 Sv. Time series of the overturning maximum (c) at 54° N and (d) considering all latitudes above 30° N, starting from the year 2000. The solid and dotted black curves represent the maximum of the streamfunction computed with the geostrophic approximation and cGENIE respectively. The red and blue curves represent the maximum of the geostrophic streamfunction when only the temperature (red) or salinity (blue) evolution is considered.





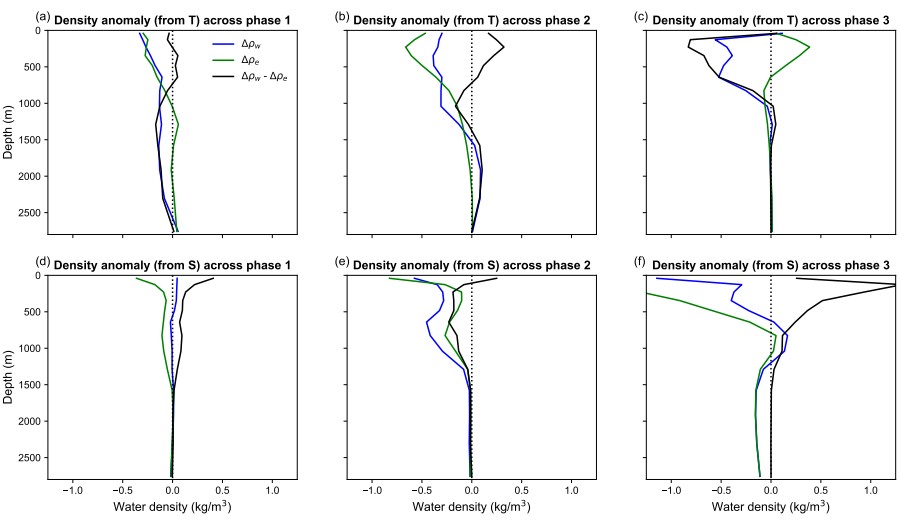

**Figure 2.** Vertical density anomaly profiles along the eastern and western Atlantic margins at 54° N for each of the three phases (2000-2100, 2100-2300 and 2300-3000). These anomalies arise either from temperature T (a-c) or salinity S changes (d-f). The blue and green curves represent the density anomaly at the eastern and western margins respectively that arises during a phase (difference between the end and the beginning of a phase). The solid black curve shows the total zonal density anomaly. The dotted black curve separates the positive from the negative density anomalies.



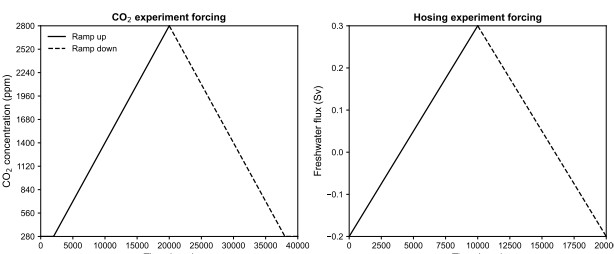

**Figure 3.** Schematic of atmospheric $CO_2$ (left) and freshwater (right) forcing for the hysteresis experiments. Solid and dashed lines represent the transient increase and decrease of the forcing, respectively.



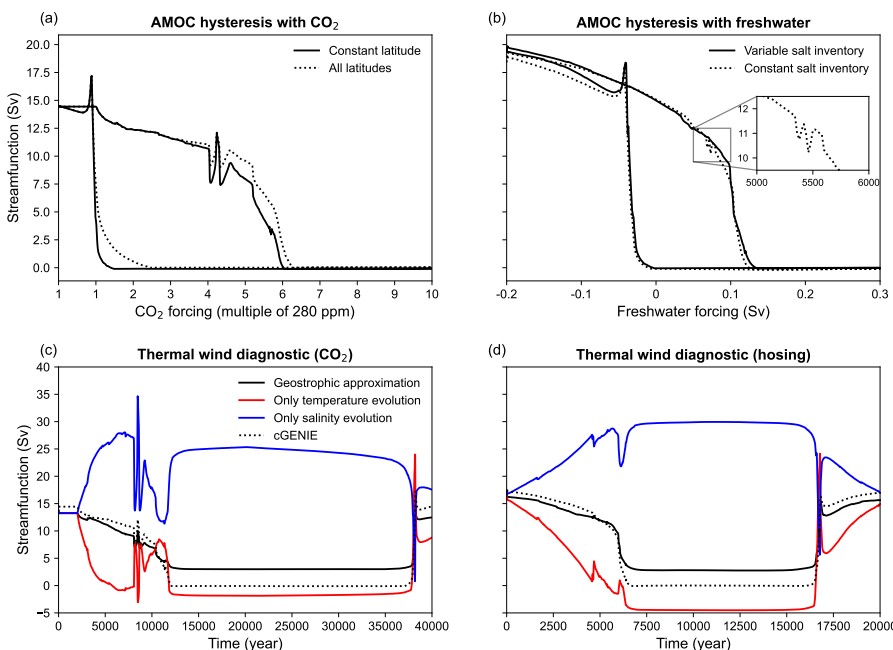

**Figure 4.** Hysteresis curves of cGENIE streamfunction maximum obtained with an (a) $CO_2$ and (b) freshwater forcing. In (a) the two black curves represent the overturning maximum at $54°$ N (solid) and considering all latitudes above $30°$ N (dotted). (b) For the freshwater forcing two different simulations were conducted, one with a constant salinity inventory (dotted black line) and another where the inventory is left variable (solid black line). The thermal wind diagnostic was performed for the (c) $CO_2$ and (d) freshwater with variable salt inventory hysteresis experiments. The evolution in (c) and (d) is expressed in years with the corresponding forcing being indicated in Figure 3.



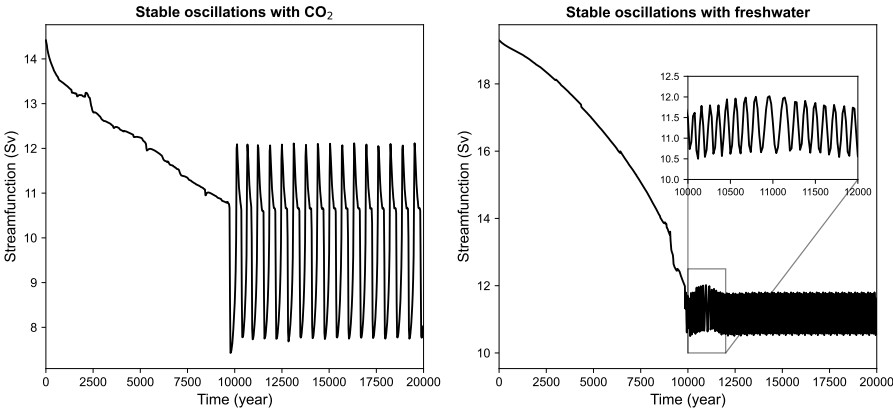

**Figure 5.** Evolution of the streamfunction maximum at 54° N when the system is forced with $CO_2$ (left) and freshwater (right) over a period of 20000 years.



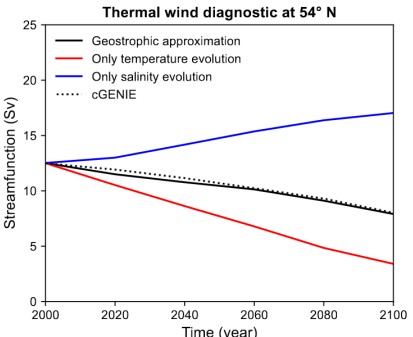
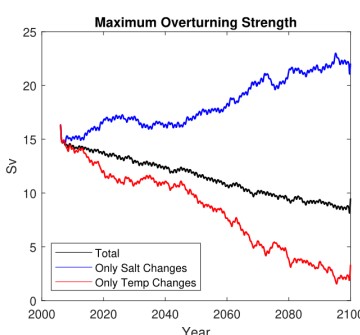

**Figure A1.** Comparison of the diagnostic obtained with cGENIE (left) and the multimodel mean made by Levang and Schmitt (2020) (right), visible on the top left panel of their Fig. 4, where colours are adapted to match our conventions.







**Figure A2.** Vertical profile of temperature (left) and salinity (right) values along the western (blue) and eastern (green) Atlantic margin at 54° N.