# Peer review of "Diagnosing the causes of AMOC slowdown in a coupled model: a cautionary tale"

_EGUsphere, 2023_

## Referee Comment (RC2)

**Review of "Diagnosing the AMOC slowdown in a coupled model: a cautionary tale".Justin Gérard and Michel Crucifix, for Earth System Dynamics.**

egusphere-2023-2001

**General comments**

This manuscript is primarily a response to a 2020 paper in the Journal of Climate on "What causes the AMOC to weaken in CMIP5?" by Samuel Levang and Raymond Schmitt. Levang and Schmitt concluded, using diagnostics based on the thermal wind relationship and zonal density gradients across the width of the basin that:

> "Therefore, in CMIP5, temperature dynamics are responsible for AMOC weakening, while freshwater forcing instead acts to strengthen the circulation in the net. These results indicate that past modelling studies of AMOC weakening, which rely on freshwater hosing in the subpolar gyre, may not be directly applicable to a more complex warming scenario."

The current manuscript uses cGENIE, an Earth System model of intermediate complexity, to test the diagnostics used by Levang and Schmitt. The principal finding is that, independent of whether reduced modellied AMOC is caused explicitly by rising $CO_2$ (temperature forcing) or hosing (freshwater forcing), the thermal wind diagnostics tend to identify changes to the temperature field as the cause of AMOC weakening.

The manuscript is generally well constructed, the results presented clearly, and appropriate conclusions drawn. While I am not an expert on Earth system models, the choice of the cGENIE model here seems well justified and appropriate. The results should be a useful addition to the field and I'm happy to recommend publication subject to authors addressing the minor comments below.

**Specific comments**

Title. Maybe "Diagnosing *the causes of* AMOC slowdown..."?

Introduction. Lines 29--31. I'm not sure about the characterisation of AMOC flows as 'clockwise' and 'anticlockwise'. I assume this is with reference to meridional sections, presented with North to the right, like those in Fig 1a,b. I think it is more understandable to describe these in terms of the location of downwelling and upwelling, and the direction of the upper and deep flows.

Results. Section 3.1 RCP8.5. Figures 1 and 2.

I found following this section, with reference to Figures 1 and 2, to be quite tricky. I think my difficult is in understanding the relationship between the density anomalies presented in Figure 2 and the maximum of the overturning streamfunction (and hence the AMOC variability) presented in Figure 1. I have some suggestions:

Line 177--179. The overturning streamfunction is not the vertical integral of the zonal density gradient, it is the vertical integral of the *zonal pressure gradient*. The pressure gradient at depth z is, in turn, the vertical integral from the surface to depth z, of the density gradient.

I think this is where my trouble is, the need to integrate twice in the vertical, in my mind, and then select a maximum, to get from the density anomaly profiles in Figure 2 to the AMOC strength in Figure 1c,d. I suggest:

- adding, either as additional panels in Fig 2 or as a new figure, the pressure anomalies in the vertical (i.e. the first vertical integral of Fig 2)
- similarly, add plots of the streamfunctions with depth (the second vertical integral of the zonal density differences).

These would help the reader understand the relationship between the density changes and the streamfunction changes.

Presenting the vertical structure of the thermal wind streamfunction diagnostic would also help to see how the temperature and salinity contributions in Fig 1c,d sum to produce the total geostrophic contribution. I assume the separate T and S contributions in Fig 1 do not simply sum to the total because the salinity and temperature streamfunction maxima appear at different depths?

Section 3.1 could possibly be usefully broken up with some subheading to aid readability

Section 3.2.

Line 271. 'in the latter case'. I think this refers to the freshwater flux hysteresis experiment, which is the former case in this sentence. Please clarify.

Lines 281+ This examination of the oscillating behaviour, while maybe interesting to Earth system modellers, seems out of place in this manuscript. I would recommend dropping it along with Fig 5.

Figure 4a,b. These each contain 4 lines (2 solid, 2 dotted). I think the text and caption only refer to two of these lines? Either delete the two which are not used, or, if they are mentioned somewhere and I've missed it, please differentiate the two pairs, maybe with different colours.

Figures A1 and A2, I'd recommend inclusion of these in the main body of the work.

Finally, a question about the use of depth-space, rather than density-space AMOC diagnostics. Particularly as far north as 54N where much of the analysis is based. At 54 N in the Atlantic Ocean there is a sizeable horizontal component to the overturning circulation which perhaps makes the e-w boundary thermal wind diagnostics less appropriate here than at lower latitudes (and is the reason why the OSNAP overturning monitoring array requires many more moorings than the RAPID array). Could the authors comment on this? Is, for example, the horizontal component of the overturning less significant in the coarser resolution, relatively simple model presented?

I enjoyed reading this and at the end feel I've learnt something useful. Thanks.

Alan Fox

---

## Author Response (AR1)

**Review of "Diagnosing the AMOC slowdown in a coupled model: a cautionary tale" — Author's response**

Justin Gérard[1] and Michel Crucifix[1]

[1]Université catholique de Louvain (UCLouvain), Earth and Life Institute (ELI), Louvain-la-Neuve, Belgium

**Correspondence:** Justin Gérard (justin.gerard@uclouvain.be)

**1 Ivica Vilibić**

The paper presents a simplified modelling effort for studying millenial changes in phases of the Atlantic Meridional Overturning Circulation. Obviously, using more complex model at these simulation timescales would take enormous amount of computational resources - through the manuscript the authors justify their choices in experiments and simplifactions with respect to the full climate models. The simulations provide some new insights to the topic, that are throughoutly discussed in the manuscript. The writing style and the English is also at high level, I have no objections. The only minor item to complain is introduction of Figure A1 and Figure A2 in the middle of the manuscript, which are not treated as annex or supplementary, so I suggest to include them as regular figures in the main text.

We thank the reviewer for her straightforward and kind review. The only minor comments will be fully addressed later as it has also been raised by the second reviewer, but in short, we will add the recommended figures to the main text.

**2 Alan Fox**

This manuscript is primarily a response to a 2020 paper in the Journal of Climate on "What causes the AMOC to weaken in CMIP5?" by Samuel Levang and Raymond Schmitt. Levang and Schmitt concluded, using diagnostics based on the thermal wind relationship and zonal density gradients across the width of the basin that: "Therefore, in CMIP5, temperature dynamics are responsible for AMOC weakening, while freshwater forcing instead acts to strengthen the circulation in the net. These results indicate that past modelling studies of AMOC weakening, which rely on freshwater hosing in the subpolar gyre, may not be directly applicable to a more complex warming scenario." The current manuscript uses cGENIE, an Earth System model of intermediate complexity, to test the diagnostics used by Levang and Schmitt. The principal finding is that, independent of whether reduced modellied AMOC is caused explicitly by rising CO2 (temperature forcing) or hosing (freshwater forcing), the thermal wind diagnostics tend to identify changes to the temperature field as the cause of AMOC weakening. The manuscript is generally well constructed, the results presented clearly, and appropriate conclusions drawn. While I am not an expert on Earth system models, the choice of the cGENIE model here seems well justified and appropriate. The results should be a useful addition to the field and I'm happy to recommend publication subject to authors addressing the minor comments below.

Thank you to the reviewer for his constructive comments on the manuscript.

25 Title. Maybe "Diagnosing the causes of AMOC slowdown..."?
The comment has been taken into account and the title has been modified.

Introduction. Lines 29–31. I'm not sure about the characterisation of AMOC flows as 'clockwise' and 'anticlockwise'. I assume this is with reference to meridional sections, presented with North to the right, like those in Fig 1a,b. I think it is more understandable to describe these in terms of the location of downwelling and upwelling, and the direction of the upper and
30 deep flows.
To clarify the meaning of the sentence, the text was adapted in the following way:
Lines 29–31: It is widely known (e.g. Dijkstra (2005)) that the atmospheric buoyancy fluxes have opposite contributions to the AMOC: a purely thermal circulation would have a downwelling flow in the North that follows the sinking of colder water in the North Atlantic, with upper and deep flows propagating towards the North Polar and equatorial regions respectively. On the
35 contrary, a strictly haline circulation would have a downwelling flow located in the equatorial zone due to the sinking of saltier water at the equator, with upper and deep flows propagating towards the equatorial and North Polar regions respectively.
This now corresponds to Lines 29–33 in the new version.

Line 177–179. The overturning streamfunction is not the vertical integral of the zonal density gradient, it is the vertical integral of the zonal pressure gradient. The pressure gradient at depth z is, in turn, the vertical integral from the surface to depth z, of
40 the density gradient.
We propose the following modification of Lines 177–179, where we chose to work with the geostrophic velocity anomaly for the first integral of the zonal density gradient as it is directly introduced in the text while the pressure gradient is not.
Lines 177—179: In equation 2, the overturning streamfunction is a vertical integral of the geostrophic velocity —following the principle of the thermal wind balance— down to the ocean sea floor. The geostrophic velocity at depth $z$ is, in turn, the vertical
45 integral of the zonal density gradient.
This now corresponds to Lines 181–183 in tne new version.

I think this is where my trouble is, the need to integrate twice in the vertical, in my mind, and then select a maximum, to get from the density anomaly profiles in Figure 2 to the AMOC strength in Figure 1c,d. I suggest:

- adding, either as additional panels in Fig 2 or as a new figure, the pressure anomalies in the vertical (i.e. the first vertical
50 integral of Fig 2)

- similarly, add plots of the streamfunctions with depth (the second vertical integral of the zonal density differences).

These would help the reader understand the relationship between the density changes and the streamfunction changes.
This comment adds a welcoming addition to the clarity of the text. Going from zonal density anomalies to the maximum of a streamfunction requires some intermediate steps. The two suggestions of the reviewer were implemented in the new Figure 5.

[Figure]

**Figure 5.** Vertical profiles of geostrophic velocity anomalies at $54°$ N for each of the three phases (a) 2000-2100, (b) 2100-2300 and (c) 2300-3000. Vertical profiles of different streamfunctions at $54°$ N for the year (d) 2100, (e) 2300 and (f) 3000. The solid black curve represents the portrayed quantity computed with the geostrophic approximation. The red and blue curves represent the same quantity when only the temperature (red) or salinity (blue) evolution is considered. The dashed black curve (d-f) represents the streamfunction computed with the geostrophic approximation of the initial condition (year 2000). This figure represents the relationship between the zonal density gradient (see Figure 4) and the streamfunction maximum illustrated in Figure 1c.

55  Furthermore, variations observed in Figure 5d-f are slightly more complex than the direct integral of the geostrophic velocity since the depth average has been removed as shown in equation (2) from the manuscript (Hirschi and Marotzke, 2007).

$$\Psi_{\mathrm{geo}}(z) = \int_{x_w}^{x_e} \int_{-H}^{z} \left( v_g(z) - \frac{1}{H} \int_{-H}^{0} v_g(z)\mathrm{d}z \right) \mathrm{d}z\mathrm{d}x$$

Presenting the vertical structure of the thermal wind streamfunction diagnostic would also help to see how the temperature and salinity contributions in Fig 1c,d sum to produce the total geostrophic contribution. I assume the separate T and S contributions

60  in Fig 1 do not simply sum to the total because the salinity and temperature streamfunction maxima appear at different depths? When it comes to velocity anomalies, combining the blue and red curves gives the black curve (as can be seen in Figures 5a-c). The same conclusion can be made with the streamfunction anomalies (not shown here). But, as the reviewer correctly pointed out, the separate T and S contributions in Figure 1 do not simply sum to equal the total geostrophic approximation because the salinity and temperature streamfunction maxima often appear at close but different depths (see Figures 5d-f).

65  This has been added to the caption of Figure 1 to clarify this point: Here, the temperature and salinity contributions do not exactly sum to equal the total geostrophic approximation because the maxima of their respective streamfunction (red and blue curves) often appear at close but different depths (see Figures 5d-f).

Every Figure number is taken from the revised manuscript. See Table 1 in this document for the correspondence with the old document.

70   Section 3.1 could possibly be usefully broken up with some subheading to aid readability.

We agree to split section 3.1 into two parts: 1) Thermal wind diagnostic and 2) Driving mechanisms.

Line 271. 'in the latter case'. I think this refers to the freshwater flux hysteresis experiment, which is the former case in this sentence. Please clarify

Line 271: Perhaps more surprising is that the hysteresis experiment with freshwater flux has a similar ratio between thermal and

75   haline contributions to that of the RCP8.5 (see Figure 7d), because we know that in the former case, the forcing is freshwater and not temperature.

This now corresponds to Line 274 in the new version.

Lines 281+ This examination of the oscillating behaviour, while maybe interesting to Earth system modellers, seems out of place in this manuscript. I would recommend dropping it along with Fig 5.

80   Figure 5 has been dropped as it does not add much valuable information to the text. We also propose to shorten the paragraph and add a sentence at the end to improve coherence:

Lines 281+: Now we return to the 'saw-tooth-shaped' event visible on both curves of Figure 7a. We find that this 'saw-tooth' behaviour is a furtive occurrence of a self-sustained oscillation that is triggered as the system evolves in quasi-equilibrium under particular $CO_2$ concentrations. The oscillation regime is also triggered in the freshwater forcing hysteresis experiment,

85   but the oscillation is weaker and only appears when the freshwater flux forcing in the Atlantic is compensated for in the Pacific (see zoom in Figure 7b). The occurrence of natural oscillations in the AMOC on the multicentennial/millennial timescale is a familiar phenomenon that has been observed across many different models (Sakai and Peltier, 1997; Thornalley et al., 2009; Peltier and Vettoretti, 2014; Sévellec and Fedorov, 2014; Li and Yang, 2022; Vettoretti et al., 2022). In cGENIE, some millennial oscillations have been highlighted by Keane et al. (2022) for different ranges of atmospheric $CO_2$ concentrations

90   and an idealized continental configuration. Oscillations in the THC have also been observed in cGENIE as freshwater forcing is applied to the system when close to a bifurcation point (Lenton et al., 2009). Self-sustained oscillations in the ocean can have different origins and a full investigation of the mechanism of their specific occurrence here is beyond the scope of the present study. However, these oscillations are responsible for the outstanding excursions in the diagnostic of thermal wind observed around year 8000 in Figure 7c, yet they do not alter the three distinct phases identified.

95   This now corresponds to Lines 285+ in the new version.

Figure 4a,b. These each contain 4 lines (2 solid, 2 dotted). I think the text and caption only refer to two of these lines? Either delete the two which are not used, or, if they are mentioned somewhere and I've missed it, please differentiate the two pairs, maybe with different colours.

Figures 4a and b contain only 2 hysteresis simulations each: one dotted and one solid. The 'additional' lines correspond to the

100   return path of these simulations when the forcing is inverted.

We propose to add the following sentence to the description of Figure 4 to improve clarity: Panels (a) and (b) exclusively feature two hysteresis simulations each: one dotted and one solid. Each simulation is then composed of 2 different parts: one where the forcing increases and the other where it decreases.

Figures A1 and A2, I'd recommend inclusion of these in the main body of the work.

This was recommended by both reviewers. Figures A1 and A2 have been included in the main text as Figures 2 and 3 respectively. The new vs old Figures are summarized in the following table, where 'New Figure 5' refers to Figure 5 introduced in this document.

| | |
|---|---|
| Figure 1 | Figure 1 |
| Figure 2 | Figure A1 |
| Figure 3 | Figure A2 |
| Figure 4 | Figure 2 |
| Figure 5 | New Figure 5 |
| Figure 6 | Figure 3 |
| Figure 7 | Figure 4 |

**Table 1.** Relationship between old and new Figures.

Finally, a question about the use of depth-space, rather than density-space AMOC diagnostics. Particularly as far north as 54N where much of the analysis is based. At 54 N in the Atlantic Ocean there is a sizeable horizontal component to the overturning circulation which perhaps makes the e-w boundary thermal wind diagnostics less appropriate here than at lower latitudes (and is the reason why the OSNAP overturning monitoring array requires many more moorings than the RAPID array). Could the authors comment on this? Is, for example, the horizontal component of the overturning less significant in the coarser resolution, relatively simple model presented?

To quantify the relative contribution between the horizontal and the zonal component of the overturning we used a diagnostic proposed in Weber et al. (2007). The diagnostic allows us to compute the meridional overturning component and azonal component of the oceanic freshwater transport. The results of this diagnostic at 54° N are the following:

- The dominant component consistently remains the meridional overturning. On average, the meridional component is four times larger than the azonal component.

- Over the first 100 years of the RCP8.5 simulation, the azonal component consistently is an order of magnitude less than the meridional component.

All these results justify the use of the thermal wind diagnostics at 54° N for the model employed in this paper.

**References**

Dijkstra, H. A.: Nonlinear physical oceanography: a dynamical systems approach to the large scale ocean circulation and El Niño, vol. 532, Springer, 2005.

125  Hirschi, J. and Marotzke, J.: Reconstructing the meridional overturning circulation from boundary densities and the zonal wind stress, Journal of Physical Oceanography, 37, 743–763, 2007.

Keane, A., Pohl, A., Dijkstra, H. A., and Ridgwell, A.: A simple mechanism for stable oscillations in an intermediate complexity Earth System Model, arXiv preprint arXiv:2201.07883, 2022.

Lenton, T. M., Myerscough, R. J., Marsh, R., Livina, V. N., Price, A. R., Cox, S. J., and team, G.: Using GENIE to study a tipping point in

130  the climate system, Philosophical Transactions of the Royal Society A: Mathematical, Physical and Engineering Sciences, 367, 871–884, 2009.

Li, Y. and Yang, H.: A theory for self-sustained multicentennial oscillation of the Atlantic Meridional Overturning Circulation, Journal of Climate, 35, 5883–5896, 2022.

Peltier, W. R. and Vettoretti, G.: Dansgaard-Oeschger oscillations predicted in a comprehensive model of glacial climate: A "kicked" salt

135  oscillator in the Atlantic, Geophysical Research Letters, 41, 7306–7313, 2014.

Sakai, K. and Peltier, W.: Dansgaard–Oeschger oscillations in a coupled atmosphere–ocean climate model, Journal of Climate, 10, 949–970, 1997.

Sévellec, F. and Fedorov, A. V.: Millennial variability in an idealized ocean model: predicting the AMOC regime shifts, Journal of Climate, 27, 3551–3564, 2014.

140  Thornalley, D. J., Elderfield, H., and McCave, I. N.: Holocene oscillations in temperature and salinity of the surface subpolar North Atlantic, Nature, 457, 711–714, 2009.

Vettoretti, G., Ditlevsen, P., Jochum, M., and Rasmussen, S. O.: Atmospheric CO2 control of spontaneous millennial-scale ice age climate oscillations, Nature Geoscience, 15, 300–306, 2022.

Weber, S., Drijfhout, S., Abe-Ouchi, A., Crucifix, M., Eby, M., Ganopolski, A., Murakami, S., Otto-Bliesner, B., and Peltier, W.: The modern

145  and glacial overturning circulation in the Atlantic ocean in PMIP coupled model simulations, Climate of the Past, 3, 51–64, 2007.